# Components of the full blood count as risk factors for colorectal cancer detection: a systematic review protocol

Pradeep S Virdee ,[1] Shona Kirtley,[1] Leena Elhussein,[1] Peter J Watkinson ,[2] Tim A Holt,[3] Jacqueline Birks[1]

[1]Centre for Statistics in Medicine, University of Oxford, Oxford, UK
[2]Kadoorie Centre for Critical Care Research and Education, Oxford University Hospitals NHS Trust, Oxford, UK
[3]Department of Primary Care Health Sciences, Oxford University, Oxford, UK

**Correspondence to**
Pradeep S Virdee;
pradeep.virdee@csm.ox.ac.uk

## ABSTRACT

**Introduction** Colorectal cancer is the fourth most common type of cancer and the second most common cause of cancer-related deaths in the UK. The full blood count (FBC) is a blood test that may play a role in early detection of the disease. Previous studies have aimed to identify how levels of individual components, such as haemoglobin, can be used to assist the diagnosis. We aim to systematically review studies to identify whether components of the FBC are risk factors for diagnosis of colorectal cancer, critically appraise the methods used to assess the association and assess performance of the components.

**Methods and analysis** The MEDLINE (via OVID), EMBASE (via OVID), CINAHL (via EBSCOhost) and Web of Science databases will be searched to identify studies reporting the association between the levels of at least one FBC component and the risk of a future diagnosis of colorectal cancer in undiagnosed individuals. Clincialtrials.gov and the WHO registry will be searched to identify relevant ongoing research. Search terms will include relevant Medical Subject Headings and Emtree headings, and free-text terms relating to FBC, colorectal cancer and diagnosis. No date or language restrictions will be applied. Two reviewers will independently identify the studies for inclusion and perform data extraction. Time intervals between the blood tests and diagnosis will form the subgroups for analysis.

**Ethics and dissemination** There is no direct patient involvement and only published articles will be reviewed; no ethical approval is required. Results from this review will set a foundation for intended future work on developing a new risk score for early detection of colorectal cancer, derived using FBC data. This systematic review will also provide guidance on the analysis of time to diagnosis. The model will be freely available to UK primary care practices.

**PROSPERO registration number** CRD42019134400.

## Strengths and limitations of this study

► The first systematic review to identify the role of components of the full blood count (FBC) from a blood test in the detection of colorectal cancer.
► As the number of studies reporting on the association between components of FBC and diagnosis of colorectal cancer is increasing over time, this review is timely.
► This systematic review will make recommendations for the development of intended future risk scores for early detection of colorectal cancer, derived using FBC data.
► This review will be limited to examining published articles; so the use of blood count values in the diagnosis of colorectal cancer as per local practice policy will not be included, unless published.

years before becoming malignant. It often goes unnoticed until patients start showing symptoms, such as abdominal pain, weight loss and change in bowel habits. At this point, the cancer has usually developed to a stage where it is difficult to treat and cannot be surgically removed.[3] The stage at diagnosis heavily influences survival. The 5-year survival is 95% if the cancer is diagnosed at stage I where the cancer is confined to the bowel lining, but 7% if at stage IV, where the cancer has spread to other organs.[4]

Patients with colorectal cancer respond well to existing interventions, such as surgery, chemotherapy and radiotherapy, if the cancer is diagnosed at an early stage. Currently, over 50% of patients are diagnosed with late-stage cancer (stages III and IV), with approximately half of these having metastases at diagnosis, compared with the earlier stages (stages I and II). Their outcomes, including survival, are much poorer than for those diagnosed with cancer at an earlier stage.[5] There are symptom-based approaches to identify the risk of colorectal cancer, such as QCancer Colorectal, a statistical prediction model widely used in UK primary care practices.[6]

## INTRODUCTION

Colorectal cancer is the fourth most common type of cancer in the UK, with around 41 800 new cases diagnosed in 2015.[1] It is the second most common cause of cancer-related death in the UK, with around 16 400 deaths in 2016.[2] Colorectal cancer develops slowly from precancerous polyps that may be present for

However, approaches to detect cancer earlier, before overt symptoms appear, would be of considerable benefit. Early detection and removal of polyps can prevent colorectal cancer from developing and improve survival.

A full blood count (FBC) is a common blood test ordered by a doctor in both primary and secondary care, because abnormalities could relate to a wide range of diseases and conditions used in clinical practices. A FBC includes up to 20 blood components (red blood cells, white blood cells, mean platelet volume, haemoglobin, haematocrit, mean corpuscular volume, mean corpuscular haemoglobin, mean corpuscular haemoglobin concentration, red blood cell distribution width, platelet, basophil #, basophil %, eosinophil #, eosinophil %, lymphocyte #, lymphocyte %, monocyte #, monocyte %, neutrophil # and neutrophil %).

It is known that the FBC may play a role in the diagnosis of colorectal cancer, with subtle changes in FBC occurring when the cancer is at a relatively early stage. Some studies have explored the relationship between the levels of specific components of the FBC and colorectal cancer diagnosis.[7 8] For example, anaemia in FBC data from UK primary care predicts the risk of colorectal cancer and iron deficiency is an independent risk factor.[8] In the last few years, a number of individual studies have reported on the association between the components of the FBC, including haemoglobin, platelet count and red cell distribution width, and diagnosis of colorectal cancer.[9-13] Furthermore, risk scores have recently been developed using methods that incorporate FBC data as predictors of colorectal cancer diagnosis. This includes the statistical prediction model, QCancer Colorectal,[6] which uses haemoglobin level as a predictor for risk of diagnosis and other algorithms, such as the ColonFlag, an Israeli risk score derived recently from FBCs using machine-learning methods.[14]

The aim of this review is to identify components of FBC as potential risk factors for colorectal cancer diagnosis, given the increasing interest in the use of FBC for early detection and to inform the development and validation of future risk scores for early detection of colorectal cancer.

### Existing systematic reviews
Systematic reviews of colorectal cancer detection already exist, four of which have been identified as reviews of blood-based characteristics. However, none of these reviews focused on the FBC blood test.

Bhardwaj et al[15] and Shah et al[16] focused on reviewing parts of the blood, such as blood-based proteins, DNA biomarkers and gene expressions that are not part of FBC. Nikolou et al[17] reviewed blood markers that were divided into four groups, nucleic acids, cytokines, antibodies and proteins, which are components of the blood and not part of FBC. Del Giudice et al[18] reviewed clinical features of suspected characteristics, but this consisted primarily of symptoms reported in the clinic. Their findings include the predictive performance of haemoglobin,

but they do not report on the remaining blood components that form the FBC, including up to 19 other components, which are also our main interest here. Usher-Smith et al[19] performed a comprehensive appraisal of prediction models for risk of colorectal cancer. However, these were not necessarily models containing components of the FBC as risk factors. Additionally, other types of studies were excluded from their review, such as retrospective cohort and case–control studies, which explored the association between components of the FBC and risk of colorectal cancer diagnosis.

Other reviews of colorectal cancer detection have a focus different from the one described here. These include Astin et al,[20] who reviewed the diagnostic value of symptoms, Williams et al,[21] who focused solely on the prediction models for patients who reported symptoms and Vega et al,[22] who summarised the available evidence concerning the diagnostic process, its pitfalls and opportunities.

To our knowledge, there are no existing reviews identifying components of the FBC as risk factors of colorectal cancer diagnosis.

### Research aims
In this review, we aim to identify and appraise studies that explored the association between the levels of blood components from FBC and the risk of diagnosis of colorectal cancer. We aim to:
► Identify which specific blood component(s) was investigated and whether it was considered associated with risk of diagnosis of colorectal cancer.
► Describe the study design and methods used to assess the association: for example, whether statistical methods were used, whether blood levels were modelled as categorical or continuous, how missing data were dealt with and the timeframe to diagnosis.
► Describe the study population and setting: for example, whether the study used electronic health data, primary or secondary care patients and symptomatic or asymptomatic patients.
► Identify whether the values of the components of FBC differ between those diagnosed with colorectal cancer and those not diagnosed.
► Studies reporting the development, validation or testing of a risk score that uses FBC component(s) for colorectal cancer diagnosis, such as prediction models, will also be included. For these studies, the following information will be collected, in addition to the points above:
► The weight the blood component(s) has on the risk of diagnosis of colorectal cancer. For example, this could be the coefficient from a regression model.
► Identify risk factors, in addition to the components of FBC, used to derive risk scores.
► Report which risk scores have undergone validation, whether it was internal and/or external and how well the risk scores performed.

## METHODS

This systematic review protocol follows the Preferred Reporting Items for Systematic review and Meta-Analysis Protocols (PRISMA-P) guidelines.[23] This review protocol was registered with the International Prospective Register of Systematic Reviews database on 13 May 2019. Screening of studies is expected to commence in August 2019. The review will be reported in accordance with the PRISMA checklist.[24]

### Participants

All adult populations (aged 18 years or above) will be considered, with the exception of those who are studied postdiagnosis of colorectal cancer.

### Search strategy

The MEDLINE (via OVID), EMBASE (via OVID), CINAHL (via EBSCOhost) and Web of Science databases will be searched to identify publications in the medical literature that report the association between components of the FBC and diagnosis of colorectal cancer. Additionally, clincialtrials.gov and the WHO registry will be searched to identify any relevant ongoing research. Search terms will include relevant Medical Subject Headings and Emtree headings, and free-text search terms, which will be searched for in the title, abstract and keyword fields. Free-text search terms will include synonyms and related variants of blood count, such as "platelet" or "basophil", colorectal cancer-related terms, such as "bowel" or "rectal" and detection related terms such as "diagnosis" or "prediction". There will be no date or language restrictions applied to the search. The proposed full search strategy for the MEDLINE database can be found in online supplementary file 1.

### Study selection

#### Data management

The results from each of the database searches will be downloaded into Rayyan,[25] a web-tool for systematic reviews and the full results set deduplicated and screened. Screening of each publication will be performed independently by two reviewers. Each reviewer will read titles and abstracts to identify the study sample for analysis using prespecified selection criteria. Disagreements between the two reviewers will be discussed until an agreement is reached. In the event that no agreement is reached, a third reviewer will be consulted for adjudication. The search results and study selection process will be reported using a PRISMA flow diagram.[24]

#### Selection criteria

We will include any primary research publication reporting the association between the value of at least one of the individual components of the FBC and the risk of a future diagnosis of colorectal cancer in undiagnosed individuals. For studies reporting the development and/or validation of a risk score, these will only be included if the risk score was derived using at least two risk factors, with at least one being a component of FBC.

We will exclude abstracts and conference proceedings, as they are likely to produce incomplete data for a thorough review. As the main interest is in the recorded value of a blood component from FBC, studies not investigating the value itself will be excluded, such as studies exploring the morphology of individual blood components. Studies using a cross-sectional design will be excluded as the data reflects a 'snapshot' at a certain time, and, hence, cannot predict the future risk. Clinical trials will be excluded as our interest is in the FBC data, not the intervention. Existing systematic reviews, correspondence and case studies pertaining to single individuals will be excluded.

### Data extraction

#### Data management

Data extracted from the publications included for analysis is intended to be recorded in the Research Electronic Data Capture electronic tool.[26] Two reviewers will read the full-text publications and independently extract data from each study. Disagreements between the two reviewers will be resolved by a third reviewer.

#### Data items

Data extracted will include items from the PRISMA checklist.[24] Data specific to this study will be extracted using a self-developed data extraction form. This form will be piloted on a small number of publications in the final sample; the publications will be selected randomly and the number chosen for piloting will depend on the number of publications included in the sample for analysis. The study aims at specifying the data items for extraction, and will include the following:

► Study characteristics (such as authors, year of publication, sample size).
► Study design (such as prospective, cohort, case–control).
► Study setting (such as geographical location, primary or secondary care, electronic health record data).
► Population characteristics (such as age, gender, symptomatic/asymptomatic).
► Outcome details (such as number of cases/controls, timeframe).
► Blood component details (such as which component was analysed, blood levels used, association of the blood component with risk of colorectal cancer).
► Methods applied to assess the association (such as how missing data were handled, whether blood levels were categorised, statistical methods used).
► Differences between those with and without a diagnosis of colorectal cancer, where possible (such as, whether blood levels differ between the two groups).

For the subset of studies reporting the development, validation or testing of a risk score for colorectal cancer diagnosis that incorporates FBC data, the following data items will also be collected:

► Weight of blood component(s) on risk (such as model coefficient, 95% CI).

 

► Risk factors used in addition to the blood component(s) (such as age, gender).
► Measures of discrimination and calibration (such as sensitivity, specificity).

## Data analysis and synthesis
### Missing data
We will contact the publication's authors requesting for the missing or incomplete data. If the data are not obtained, we will provide a description of the missing data for each study.

### Analysis methods
Quantitative data will be summarised using descriptive statistics, such as means with SD or medians with interquartile range for continuous data, and counts with proportions for categorical data and narrative synthesis. Summaries will also be provided in graphical and tabular forms.

Where appropriate and feasible, we will use random-effects meta-analysis and forest plots to pool data across studies, with the $I^2$ statistic to assess heterogeneity. The application of statistical methods for analysis will be informed by the number of publications with common measures. For example, if many case–control studies report effect estimates, such as ORs or risk ratios of blood component levels between those with and without the diagnosis of colorectal cancer, the effect estimates will be pooled using a random-effects meta-analysis. For all statistical analyses, a two-sided 5% significance level will be used.

### Subgroups for analysis
The time period between the blood test and diagnosis of colorectal cancer will be collected, such as whether the studies report a 2-year or 5-year risk of diagnosis. It is intended that time intervals will be separated into bands, such as 6-monthly or yearly intervals, before diagnosis, forming subgroups for analysis. The analyses will be performed for each time band separately. These results will set a foundation for intended future work on the analyses of time-to-diagnosis.

Planned future work will use findings from this review to develop a new risk score for early detection of colorectal cancer in the UK. Hence, studies performed in the UK population will form a subgroup for analysis.

### Assessment of bias
Risk of bias, such as analysis bias and publication bias, in each study will be assessed using tools appropriate to the design of the study. For example, the Quality In Prognosis Studies tool[27] may be used for the studies of associations and the Cochrane Prediction model Risk Of Bias ASsessment Tool)[28] may be used for the subsets identified as prediction modelling studies. Studies considered to have a high risk of bias will be excluded in a sensitivity analysis.

### Ethics and dissemination
Ethical approval is not required for this systematic review as there will be no direct patient investigations in this study and only published articles will be systematically reviewed. Results from this systematic review will be published in an open access journal.

### Patient and public involvement
Patients and public reviewed this protocol and will review the results to apply a patient perspective to the research.

## DISCUSSION
Many studies have explored the role that individual components of FBC may have in the diagnosis of colorectal cancer. Such exploration has recently become popular. However, no changes have been incorporated widely in general practice to make use of such data for early detection of the disease. Despite existing methods for colorectal cancer detection, such as screening, and methods to assist detection, such as the widely-used prediction model QCancer Colorectal, lack of early detection remains a major health problem, with the majority of colorectal cancers in the UK diagnosed at a later stage.[5] The ability to identify cases of colorectal cancer at a time point earlier than that possible with existing methods would be of considerable benefit and could save many lives.

Our team has a set of FBC data available from a large cohort of patients from UK primary care practices. However, optimal strategies for modelling the data, such as handling biologically implausible blood values and analysing changes over time, are unclear. Having appraised individual studies, this review will provide guidance on the specific components to analyse, that is, those identified as risk factors, and suggest appropriate statistical modelling strategies for implementation. Intended future work includes developing and validating a new risk score for early detection of colorectal cancer, derived using statistical modelling of FBC components based on the findings of this review. This model will be made freely available to primary care practices in the UK. Primary care practices will have the opportunity to use the model to identify the likelihood of an individual being diagnosed with colorectal cancer in the future.

**Acknowledgements** The authors would like to thank Pete Wheatstone, Margaret Ogden and Julian Ashton for their input into the protocol as patient and public involvement representatives.

**Contributors** PSV, TAH and JB developed the study and research question. PSV developed this systematic review protocol. SK developed the search strategy with inputs from PSV and LE and oversight from PJW, TAH and JB. PSV developed the data extraction sheet with inputs from LE and oversight from PJW, TAH, and JB. PSV and SK will perform the final study search. PSV and LE will perform screening, data extraction and analysis under the supervision of PJW, TAH and JB. PSV drafted this protocol and all authors developed and approved the final manuscript before submission.

**Funding** This study is funded by the National Institute for Health Research Doctoral Research Fellowship programme (DRF-2018-11-ST2-057).

**Disclaimer** The funders had no role in the development of this protocol.

**Competing interests** None declared.

**Patient consent for publication** Not required.

**Provenance and peer review** Not commissioned; externally peer reviewed.

**ORCID iDs**
Pradeep S Virdee http://orcid.org/0000-0002-3006-8730
Peter J Watkinson http://orcid.org/0000-0003-1023-3927

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
