## [Reviewer comments · BMJ Open]

ARTICLE DETAILS

TITLE (PROVISIONAL)	Components of the full blood count as risk factors for colorectal cancer detection: a systematic review protocol
AUTHORS	Virdee, Pradeep; Kirtley, Shona; Elhussein, Leena; Watkinson, Peter; Holt, Tim; Birks, Jacqueline

VERSION 1 – REVIEW

REVIEWER	Paschalis Gavriilidis Imperial College Healthcare NHS Trust Hammersmith Hospital Department of HPB and Oesophagogastric surgery London W12 0HS, UK
REVIEW RETURNED	17-Aug-2019

GENERAL COMMENTS	It was my pleasure to review the manuscript. The idea to explore the role of the individual components of FBC in early diagnosis of colorectal cancer and consequently to develop and validate a new risk score could be of considerable benefit.
---

REVIEWER	Saeed Dastgiri Department of Family and Community Medicine Tabriz University of Medical Sciences Tabriz, Post Code: 5166615739, Iran
REVIEW RETURNED	27-Sep-2019

GENERAL COMMENTS	This is a systematic review protocol aiming to investigate whether components of the full blood count are risk factors for diagnosis of colorectal cancer. The manuscript is well written/organised. It is acceptable for publication in BMJ open. However, I would suggest the authors to add the "implications" of the findings of this review (once performed) in the manuscript, and in the abstract.
---

REVIEWER	Michele Navarra University of Messina, Italy
REVIEW RETURNED	05-Nov-2019

GENERAL COMMENTS	The paper "Components of the full blood count as risk factors for colorectal cancer detection: a systematic review protocol", by Virdee and co-workers, consists in a protocol of systematic review aimed to identify and appraise studies that explored the association between levels of blood components from a full blood count and risk of diagnosis of colorectal cancer. The study is well designed and there are enough details to instil confidence that the study will be conducted and analysed properly.
---

VERSION 1 – AUTHOR RESPONSE

Reviewer: 1

Comment: The idea to explore the role of the individual components of FBC in early diagnosis of colorectal cancer and consequently to develop and validate a new risk score could be of considerable benefit.

Response: We thank you for your feedback. It is very much appreciated.

Reviewer: 2

Comment: This is a systematic review protocol aiming to investigate whether components of the full blood count are risk factors for diagnosis of colorectal cancer. The manuscript is well written/organised. It is acceptable for publication in BMJ open. However, I would suggest the authors to add the "implications" of the findings of this review (once performed) in the manuscript, and in the abstract.

Response: Thank you for your suggestion. The manuscript already discusses some implications of this systematic review in both the Discussion and Abstract. However, we recognise these implications are statistical and have updated the manuscript accordingly to discuss clinical implications. In the Abstract, under Ethics and dissemination, we have added the following text as the last sentence: "The model will be freely available to UK primary care practices."

In the Discussion, we have also added text to the end of the last sentence. The last sentence now reads:

"Intended future work includes developing and validating a new risk score for early detection of colorectal cancer, derived using statistical modelling of FBC components, based off the findings of this review. This model will be made freely available to primary care practices in the UK. Primary care practices will have the opportunity to use the model to identify the likelihood of an individual being diagnosed with colorectal cancer in the future."

Reviewer: 3

The paper "Components of the full blood count as risk factors for colorectal cancer detection: a systematic review protocol", by Virdee and co-workers, consists in a protocol of systematic review aimed to identify and appraise studies that explored the association between levels of blood components from a full blood count and risk of diagnosis of colorectal cancer.

The study is well designed and there are enough details to instil confidence that the study will be conducted and analysed properly.

Response: We thank you for your comment and appreciate your feedback.